# ON NONDETERMINISM AND INSTABILITY IN NEURAL NETWORK OPTIMIZATION

## ABSTRACT

Optimization nondeterminism causes uncertainty when improving neural networks, with small changes in performance difficult to discern from run-to-run variability. While uncertainty can be reduced by training multiple copies of a model with different random seeds, doing so is time-consuming, costly, and makes reproducibility challenging. Despite this, little attention has been paid towards establishing an understanding of this problem. In this work, we establish an experimental protocol for understanding the effect of optimization nondeterminism on model diversity, which allows us to study the independent effects of a variety of sources of nondeterminism. Surprisingly, we find that each source of nondeterminism all have similar effects on multiple measures of model diversity. To explain this intriguing fact, we examine and identify the instability of model training, when taken as an end-to-end procedure, as the key determinant. We show that even one-bit changes in initial model parameters result in models that converge to vastly different values. Last, we demonstrate that recent methods in accelerated model ensembling hold promise for reducing the effects of instability on run-to-run variability.

## 1 INTRODUCTION

Consider this common scenario: you have a baseline "current best" model, and are trying to improve it. Now, one of your experiments has produced a model whose metrics are slightly better than the baseline. Yet you have your reservations — how do you know the improvement is "real", and not due to random fluctuations that create run-to-run variability?

Similarly, consider performing hyperparameter optimization, in which there are many possible values for a set of hyperparameters, and you find minor differences in performance between them. How do you pick the best hyperparameters, and how can you be sure that you've actually picked wisely?

In both scenarios, the standard practice is to perform multiple independent training runs of your model to understand its variability. While this does indeed help address the problem, it can be extremely wasteful, increasing the time required for effective research, using more computing power, and making reproducibility more difficult, while still leaving some uncertainty.

Ultimately, the source of this problem is the nondeterminism in optimizing models — randomized components of model training that cause each training run to produce different models with their own performance characteristics. Nondeterminism itself occurs due to many factors: while the most salient source is the random initialization of parameters, other sources exist, including random shuffling of training data, per-example stochasticity of data augmentation, any explicit random operations (e.g. dropout (Srivastava et al., 2014)), asynchronous model training (Recht et al., 2011), and even nondeterminism in low-level libraries such as cuDNN (Chetlur et al., 2014), which are present to improve throughput on hardware accelerators.

Despite the clear impact nondeterminism has on the efficacy of modeling, relatively little attention has been paid towards understanding its mechanisms, even in the classical supervised setting. In this work, we establish an experimental protocol for analyzing the impact of nondeterminism in model training, allowing us to quantify the independent effect of each source of nondeterminism. In doing so, we make a surprising discovery: each source has nearly the same effect on the variability of final model performance. Further, we find each source produces models of similar diversity, as measured by correlations between model predictions, functional changes in model performance while ensembling, and state-of-the-art methods of model similarity (Kornblith et al., 2019). To emphasize

one particularly interesting result: nondeterminism in low-level libraries like cuDNN can matter just as much with respect to model diversity and variability as varying the entire network initialization.

We explain this mystery by demonstrating that it can be attributed to an inherent numerical *instability* in optimizing neural networks — when training with SGD-like approaches, we show that small changes to initial parameters result in large changes to final parameter values. In fact, the instabilities in the optimization process are extreme: *changing a single weight by the smallest possible amount within machine precision* ($\sim 6 * 10^{-11}$) *produces nearly as much variability as all other sources combined*. Therefore, any source of nondeterminism that has any effect at all on model weights is doomed to inherit at least this level of variability.

Last, we present promising results in reducing the effects of instability. While we find that many approaches result in no apparent change, we demonstrate that methods for accelerated model ensembling actually do reduce the variability of trained models without an increase in model training time, providing the first encouraging signs for tractability of the problem.

## 2 RELATED WORK

NONDETERMINISM.    Relatively little prior work has studied the effects of nondeterminism on model optimization. Within reinforcement learning, nondeterminism is recognized as a significant barrier to reproducibility and evaluating progress in the field (Nagarajan et al., 2018; Henderson et al., 2018; Islam et al., 2017; Machado et al., 2018). In the setting of supervised learning, though, the focus of this work, the problem is much less studied. Madhyastha & Jain (2019) aggregate all sources of nondeterminism together into a single random seed and analyze the variability of model attention and accuracy as a function of it across various NLP datasets. They also propose a method for reducing this variability (see Sec. A for details of our reproduction attempt). More common in the field, results across multiple random seeds are reported (see Erhan et al. (2010) for a particularly extensive example), but the precise nature of nondeterminism's influence on variability goes unstudied.

INSTABILITY.    We use the term "stability" to refer to numerical stability, in which a stable algorithm is one for which the final output (converged model) does not vary much as the input (initial parameters) are changed. Historically, the term "stability" has been used both in learning theory (Bousquet & Elisseeff, 2002), in reference to vanishing and exploding gradients (Haber & Ruthotto, 2017), and in the adversarial robustness community for a particular form of training (Zheng et al., 2016).

## 3 NONDETERMINISM

Many sources of nondeterminism exist when optimizing neural networks, each of which can affect the variability and performance of trained models. We begin with a very brief overview:

PARAMETER INITIALIZATION.    When training a model, parameters without preset values are initialized randomly according to a given distribution, *e.g.* a Gaussian with mean 0 and variance determined by the number of input connections to the layer (Glorot & Bengio, 2010; He et al., 2015).

DATA SHUFFLING.    In stochastic gradient descent (SGD), the overall gradient is approximated by the gradient on a random subset of examples. Most commonly, this is implemented by shuffling the training data, after which the data is iterated through in order. Shuffling may happen either once, before training, or in between each epoch of training, the variant we use in this work.

DATA AUGMENTATION.    A very common practice, data augmentation refers to randomly altering each training example to artificially expand the training dataset. For example, in the case of images, it is common to randomly flip an image, which encourages invariance to left/right orientation.

STOCHASTIC REGULARIZATION.    Some forms of regularization, such as Dropout (Srivastava et al., 2014), take the form of stochastic operations in a model during training. Dropout is the most common instance of this type of regularization, with a variety of others also in relatively common use, such as DropConnect (Wan et al., 2013), variational dropout (Gal & Ghahramani, 2016), and variable length backpropagation through time (Merity et al., 2017), among many others.

LOW-LEVEL OPERATIONS.    An underlooked source of nondeterminism, the very libraries that many deep learning frameworks are built on, such as cuDNN (Chetlur et al., 2014) often are run

nondeterministically for performance reasons. This nondeterminism is small in magnitude — in one test we performed this caused a difference of roughly $0.003\%$. In the case of cuDNN, the library we test with, it is possible to disable nondeterministic behavior, incurring a speed penalty typically on the order of $\sim 15\%$. However, unlike the other sources of nondeterminism, it is not possible to "seed" this nondeterminism; it is only possible to turn it on or off, but not control its nondeterministic behavior.

### 3.1 PROTOCOL FOR TESTING EFFECTS OF NONDETERMINISM

DIVERSITY IN PERFORMANCE. Our protocol for testing the effects of sources of nondeterminism is based on properly controlling for each source. In general, suppose there are $N$ sources of nondeterminism, with source $i$ controlled by a seed $S_i$. To test the effect of source $i$, we keep all values $S_j, j \neq i$ set to a constant, and vary $S_i$ with $R$ different values, where $R$ is the number of independent training runs performed. For sources of nondeterminism which cannot be effectively seeded, such as cuDNN, we indicate one of these values as the deterministic value, which it must be set to when varying the other sources of nondeterminism.

For example, suppose that we wish to study three sources of nondeterminism, denoting $S_1$ the seed for random parameter initialization, $S_2$ for training data shuffling, and $S_3$ for cuDNN, where $S_3 = 1$ is the deterministic value for cuDNN. To test the effect of random parameter initialization, with a budget of $R = 30$ training runs, then we set $S_3$ to the deterministic value of 1, $S_2$ to an arbitrary constant (also 1 for simplicity), and test 30 different values of $S_1$. All together, this corresponds to training models for $(S_1, S_2, S_3) \in \{(i, 1, 1)\}_{i=1}^{30}$, producing a set of 30 models. To look at variability according to a particular evaluation metric (*e.g.* cross-entropy or accuracy for classification), we calculate the standard deviation (across all $R = 30$ models) of the metric. Note that it is also possible to test the effects of several sources of nondeterminism in tandem this way. For example, to test all sources of nondeterminism together, the set of models can be changed to $(S_1, S_2, S_3) \in \{(i, i, 0)\}_{i=1}^{R}$.

DIVERSITY IN REPRESENTATION. Beyond looking at diversity of test set generalization, though, it is worth examining how different the representations of trained models actually are — even though the diversity in performance might be similar between models trained with different types of nondeterminism, it might be the case that one type of nondeterminism produces models that have learned largely similar concepts, with the variance in generalization due to other factors. In order to rigorously examine these, we consider four distinct analyses on the functional behavior of models:

The first and most straightforward metric we consider is the average disagreement between pairs of models, where higher disagreement corresponds to higher levels of diversity. In contrast to our other metrics, this considers only the argmax of a model's predictions, which makes it both the most limited and the most interpretable of the group. This metric has also been used recently to compare network similarity in the context of network ensembles (Fort et al., 2019).

Second, we consider the average correlation between the predictions of two models, *i.e.* the expectation (across pairs of models from the same nondeterminism source), of the correlation of predictions, calculated across examples and classes. For example, for a classification problem, the predicted logits from each of $R$ models are flattened into vectors of length $N * C$ (with $N$ the number of test examples and $C$ the number of classes), and we calculate the mean correlation coefficient of the predictions across all $\binom{R}{2}$ pairs of models. We use Spearman's $\rho$ for the correlation coefficient, but note that others such as Pearson's $r$ are possible and yield similar conclusions. For this metric, a lower score indicates a more diverse set of models.

The third analysis we perform examines the change in performance from ensembling two models from the same source of nondeterminism. The intuition is as follows: If a pair of models are completely redundant, then ensembling them would result in no change in performance. However, if models actually learn different representations, then we expect an improvement from ensembling, with a greater improvement the greater the diversity in a set of models. Denoting by $f(S_i)$ some particular evaluation metric $f$ calculated on the predictions of model $S_i$, this change is equivalent to:

$$\frac{1}{\binom{R}{2}} \sum_{i=1}^{R} \sum_{j=i+1}^{R} \left( f\left(\frac{S_i + S_j}{2}\right) - \frac{f(S_i) + f(S_j)}{2} \right) \tag{1}$$

| Nondeterminism Source | Accuracy SD (%) | Cross-Entropy SD | Pairwise Disagree (%) | Pairwise Corr. | Ensemble Δ (%) |
|---|---|---|---|---|---|
| Parameter Initialization | $0.22 \pm 0.02$ | $0.0080 \pm 0.0005$ | 10.7 | 0.872 | 1.82 |
| Data Shuffling | $0.25 \pm 0.02$ | $0.0087 \pm 0.0005$ | 10.6 | 0.871 | 1.81 |
| Data Augmentation | $0.22 \pm 0.02$ | $0.0075 \pm 0.0005$ | 10.7 | 0.872 | 1.83 |
| cuDNN | $0.21 \pm 0.01$ | $0.0087 \pm 0.0007$ | 10.5 | 0.873 | 1.76 |
| Data Shuffling + cuDNN | $0.21 \pm 0.01$ | $0.0083 \pm 0.0006$ | 10.6 | 0.871 | 1.80 |
| Data Shuffling + Aug. + cuDNN | $0.21 \pm 0.01$ | $0.0077 \pm 0.0005$ | 10.7 | 0.871 | 1.84 |
| All nondeterminism sources | $0.26 \pm 0.02$ | $0.0075 \pm 0.0005$ | 10.7 | 0.871 | 1.82 |

Table 1: The effect of each source of nondeterminism and several combinations of nondeterminism sources for ResNet-14 on CIFAR-10. The second and third columns give the standard deviation of accuracy and cross-entropy across 100 runs, varying only the nondeterminism source (700 trained models total). Also given are error bars, corresponding to the standard deviation of the standard deviation. The fourth, fifth, and sixth columns give the average percentage of examples models disagree on, the average pairwise Spearman's correlation coefficient between predictions, and the average change in accuracy from ensembling two models, respectively (Sec. 3.1).

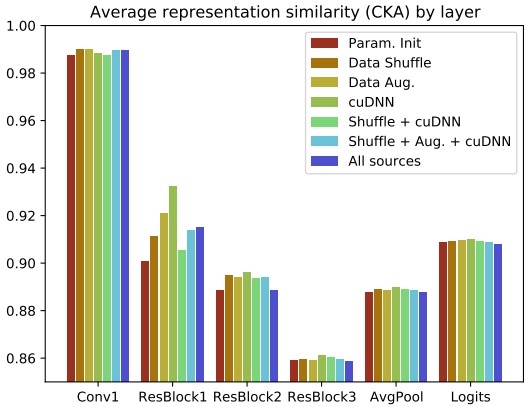

Figure 1: Average representation similarity as determined by CKA (Kornblith et al., 2019) for ResNet-14 on CIFAR-10 across nondeterminism sources. Layers are selected uniformly across the network, ranging from the first convolutional layer (Conv1), the output of each residual block (ResBlock1,2,3), the penultimate average pooling layer (AvgPool), and the logits themselves (Logits). Shown is the average CKA value between pairs of models.

Last, for a more detailed view of learned representations within a network, we consider a state-of-the-art method for measuring the similarity of learned neural network representations, centered kernel alignment (CKA) (Kornblith et al., 2019), which has been used to analyze models trained with different random initializations, widths, and even entirely different architectures. We use the linear version of CKA, which Kornblith et al. (2019) found performed similarly to an RBF kernel.

## 3.2 EXPERIMENTS IN IMAGE CLASSIFICATION

We begin our study of nondeterminism with the fundamental task of image classification. We test our protocol using CIFAR-10 (Krizhevsky et al., 2009), a 10-way classification dataset with 50,000 training images of resolution $32 \times 32$ pixels and 10,000 for testing. In these initial experiments, we use a 14-layer ResNet model (He et al., 2016), trained with a cosine learning rate decay (Loshchilov & Hutter, 2016) for 500 epochs with a maximum learning rate of $.40$, three epochs of learning rate warmup, a batch size of 512, momentum of 0.9, and weight decay of $5 \cdot 10^{-4}$, obtaining a baseline accuracy of 90.0%. Data augmentation consists of random crops and horizontal flips. All experiments were done on two NVIDIA Tesla V100 GPUs with `pytorch` (Paszke et al., 2019).

We show the results of our protocol in this setting in Table 1. What we find is surprising — while there are slight differences, each source of nondeterminism has very similar effects on the variability of final trained models, measured across both accuracy and cross-entropy. In fact, random parameter initialization, arguably the most salient form of nondeterminism, does not stand out based on any of these metrics. Even when combining multiple sources of nondeterminism, there's remarkably little difference — nothing more than one or two standard deviations compared to any individual source of nondeterminism. In short, each source of nondeterminism has roughly the same effect on the variability in final model performance, with no combination of nondeterminism sources a factor of even two more important than any other.

| Nondeterminism Source | PPL SD | Pairwise Disagree (%) | Ensemble PPL $\Delta$ |
|---|---|---|---|
| Parameter Initialization | $0.20 \pm 0.01$ | 17.3 | -2.07 |
| Stochastic Operations | $0.19 \pm 0.01$ | 17.3 | -2.08 |
| All nondeterminism sources | $0.18 \pm 0.01$ | 17.4 | -2.07 |

Table 2: The effect of each source of nondeterminism for a QRNN on Penn Treebank; 100 runs per row. Note that lower PPL is better, so changes in Ensemble PPL are negative.

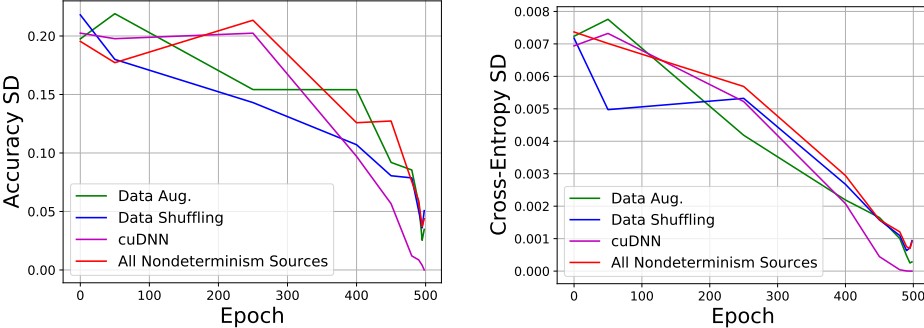

Figure 2: The effect of the onset of nondeterminism on the variability of accuracy and cross-entropy in converged models. Each point corresponds to training a set of 20 models deterministically for a certain number of epochs (x-axis), then enabling a source of nondeterminism by varying its seed starting from that epoch and continuing through the end of training.

Turning toward representational diversity on a per-layer level, average CKA values across 6 representative layers are shown in Fig. 1, done for pairwise combinations of 25 models (due to the computational requirements of CKA). Consistent with other analyses, CKA reveals that while there are some differences in representational similarity between models trained with different sources of nondeterminism, especially the output of the first residual block, by and large these differences are small, easily dwarfed in size by differences across layers.

### 3.3 EXPERIMENTS IN LANGUAGE MODELING

Here we show that this phenomenon is not unique to image classification, applying the same experimental protocol to language modeling. For these experiments, we employ a small quasi-recurrent neural network (QRNN) (Bradbury et al., 2016) on Penn Treebank (Marcus et al., 1993), using the publicly available code of Merity et al. (2017). This baseline model uses a 256-dimensional word embedding size, 512 hidden units per layer, and 3 layers of recurrent units, obtaining a perplexity (PPL) of 75.49 on the Penn Treebank test set.

For this task, two sources of nondeterminism are relevant: the random parameter initialization, and multiple stochastic operations including variations of dropout and variable length backpropagation through time, which share a common seed. To measure diversity in performance, PPL is the most widely-accepted metric, and for diversity in representation we focus on only two metrics (pairwise disagreement and benefits from ensembling) because CKA was not designed for variable-length input and common computing libraries (Virtanen et al., 2020) are not efficient enough to calculate $O(R^2)$ correlation coefficients with such large inputs. We show results in Table 2, where we find even less difference in variability between the two sources of nondeterminism considered across all three diversity metrics, showing that the phenomenon isn't unique to image classification or ResNets.

### 3.4 NONDETERMINISM THROUGHOUT TRAINING

One hypothesis for the cause of this phenomenon is that models are particularly sensitive to changes in the initial phase of learning, which recent work has shown to be the case in other contexts (Achille et al., 2019; Frankle et al., 2020). With our experimental protocol, this is straightforward to test: If this were the case, then training networks identically for the first $N$ epochs and only then introducing these sources of nondeterminism would result in a significantly lower diversity in trained models, measured across any metric of choice, *e.g.* variance in cross-entropy or accuracy. Furthermore, by varying $N$, we can actually determine *when* in training each source of nondeterminism has its effect (for sources that vary over the course of training, *i.e.* not random parameter initialization).

| Nondeterminism Source | Accuracy SD (%) | Cross-Entropy SD | Pairwise Disagree (%) | Pairwise Corr. | Ensemble Δ (%) |
|---|---|---|---|---|---|
| Random Bit Change | $0.21 \pm 0.01$ | $0.0070 \pm 0.0004$ | 10.6 | 0.874 | 1.82 |

Table 3: The effect of instability — randomly changing a single weight by one bit for ResNet-14 on CIFAR-10. Also see Table 1 for comparison.

| Nondeterminism Source | PPL SD | Pairwise Disagree (%) | Ensemble PPL Δ |
|---|---|---|---|
| Random Bit Change | $0.19 \pm 0.01$ | 17.7 | -2.07 |

Table 4: The effect of instability for a QRNN on Penn Treebank. Also see Table 2 for comparison.

We do this experiment for the ResNet-14 model on CIFAR-10 in Fig. 2, where we find that it is *not* only the beginning of training that is sensitive to changes in nondeterminism, refuting this hypothesis about the onset of the effects of nondeterminism. Instead, the effect of nondeterminism is nearly as high when enabling it even after 50 epochs (10% of the way through training), and we see only a gradual reduction in final model variability as the onset of nondeterminism is moved later and later.

## 4 INSTABILITY

Why does each source of nondeterminism have a similar effect on model diversity? We answer this question by finding the smallest possible change that produces as diverse a range of models, revealing the *instability* in optimizing neural networks.

### 4.1 INSTABILITY AND NONDETERMINISM

To demonstrate, consider again a ResNet-14 trained on CIFAR-10, with a cross-entropy loss on the test set of $0.3519$ and accuracy of $90.0\%$. Now consider another model trained in an identical fashion, with exactly equal settings for all sources of nondeterminism, but one extremely small change: we randomly pick a single weight in the first layer and change its value by the smallest amount possible in a 32-bit floating point representation, *i.e.* an addition or subtraction of a single bit in the least-significant digit. As an example, in one model this changed a value from approximately $-0.0066514308$ to $-0.0066514313$, a difference on the order of $5 \cdot 10^{-10}$.

What happens when we optimize this model, which differs from the original only by a miniscule amount, in a single weight, itself within a single layer? By the end of the first epoch of training, still in a phase where the learning rate is warming up, the new model already differs in accuracy by $0.18\%$ compared to the original ($25.74\%$ vs $25.56\%$). With one more epoch of training the difference is a larger $2.33\%$ ($33.45\%$ vs $31.12\%$), and by the end of the third epoch, as the learning rate warmup has just ended, the difference is a staggering $10.42\%$ ($41.27\%$ vs $30.85\%$). Finally, at the end of training, the differences shrink again, with the new model converged to an accuracy of $90.12\%$ and a cross-entropy of $0.34335$. When viewing the optimization process end-to-end, with the initial parameters as the input and a given performance metric as the output, this implies a condition number $\frac{\|\delta f\|}{\|\delta x\|}$ of at least $1.8 \cdot 10^7$ for cross-entropy and $2.6 \cdot 10^8$ for accuracy.

To identify whether instability is the primary reason for why each source of nondeterminism appears equally important, we can follow our testing protocol from Sec. 3 — this time, our source of nondeterminism is randomly picking a different weight to change in each model training run, then either incrementing or decrementing it to the next available floating-point value. We show the results in Table 3 for image classification on CIFAR-10 (*c.f.* Table 1 for comparison) and Table 4 for language modeling on Penn Treebank (*c.f.* Table 2), where we find that even this miniscule change produces roughly as much variability in model performance as every other source of nondeterminism.

From this, it is easy to see why every other source of nondeterminism has similar effects — so long as a nondeterminism source produces any change in model weights, whether by changing the input slightly, altering the gradient in some way, or any other effect, it will incur *at least* as much model variability as caused by instability of model optimization.

| Ensemble | Accuracy SD (%) | Cross-Entropy SD | Pairwise Disagree (%) | Pairwise Corr. | Ensemble $\Delta$ (%) |
|---|---|---|---|---|---|
| Snapshot | $0.19 \pm 0.02$ | $0.0044 \pm 0.0003$ | 6.1 | 0.957 | 0.63 |
| Vanilla ($N = 1$) | $0.26 \pm 0.02$ | $0.0075 \pm 0.0005$ | 10.7 | 0.871 | 1.82 |
| Vanilla ($N = 2$) | $0.19 \pm 0.02$ | $0.0044 \pm 0.0004$ | 6.9 | 0.929 | 0.89 |
| Vanilla ($N = 3$) | $0.15 \pm 0.02$ | $0.0033 \pm 0.0005$ | 5.5 | 0.951 | 0.59 |
| Vanilla ($N = 4$) | $0.17 \pm 0.02$ | $0.0030 \pm 0.0004$ | 4.6 | 0.963 | 0.43 |
| Vanilla ($N = 5$) | $0.12 \pm 0.02$ | $0.0028 \pm 0.0004$ | 4.1 | 0.970 | 0.34 |

Table 5: Snapshot ensembles compared with regular ensembles for ResNet-14 on CIFAR-10, with all nondeterminism sources enabled. $N$ denotes the number of component models in a regular ("Vanilla") ensemble. The snapshot ensemble is based on 100 runs of model training, and all vanilla ensembles are drawn from a pool of 100 independent model runs.

## 4.2 INSTABILITY IS DUE TO DEPTH

Instability is due to network depth, with any network greater than a single layer exhibiting instability.

Theoretically, a linear model, optimized with a cross-entropy and with an appropriate learning rate schedule, should always converge to a global minimum due to convexity. In practice, we find an even stronger property: when we modify the initial weights by a single bit, beyond simply converging to the same weights, the entire optimization trajectory stays close to that of an unperturbed model, never differing by more than a vanishingly small amount. At convergence, a set of linear models trained in this way with only single random bit changes had a final accuracy SD of 0 (*i.e.* no changes in any test set predictions) and cross-entropy SD of $\sim 3 \cdot 10^{-8}$, far below that of any deeper model.

In contrast, instability occurred as soon as a single hidden layer was added, with an accuracy SD of 0.28 and cross-entropy SD of 0.0055 for a model with a fully-connected hidden layer, and an accuracy SD of 0.14 and cross-entropy SD of 0.0024 when the hidden layer is convolutional, both a factor of 10,000 greater than the linear model. See Appendix Table 8 for full details and Appendix Sec. C for a visualization of the effects of instability during training.

## 5 A PATH FORWARD

Here we identify and demonstrate a family of approaches that holds promise for reducing the variability caused by nondeterminism and instability. For learnings on approaches which *did not* reduce variability, we direct the reader to Appendix Sec. A.

As previously mentioned, the standard practice for mitigating run-to-run variability is to train multiple independent copies of a model, gaining a more robust estimate for performance by considering an aggregate measure of a metric of interest (*e.g.* the mean). An alternative approach is ensembling, which shares the intuition of multiple independent training runs, but differs in that the predictions themselves are averaged, after which only the performance of the ensembled model itself is compared. By itself, ensembling does not substantially reduce the burden caused by nondeterminism and instability, since it requires the same computation as training multiple copies of models. Despite this, there's a way out: recent works investigating accelerated ways of constructing ensembles, in which only one training run is needed (Huang et al., 2017; Garipov et al., 2018; Wen et al., 2020). While these approaches typically underperform ensembles composed out of truly independent models, the nature of their accelerated training provides compelling motivation for reducing variability.

The approach we focus on is Snapshot Ensembles (Huang et al., 2017), which creates an ensemble of models during the training of a single model. It does this via usage of a cyclic learning rate schedule, where the members of the ensemble are drawn from instances during training when the learning rate goes to 0 (before rising again due to the cyclic learning rate). We compare a snapshot ensemble approach with 5 cycles in its learning rate (*i.e.* model snapshots are taken after every 100 epochs of training) to ordinary ensembling, in which multiple models are trained entirely from scratch.

Results are shown in Table 5, where we focus on CIFAR-10 with all sources of nondeterminism enabled. The snapshot ensemble we identified, despite training only a single model, had variability in

| Nondeterminism Source | Accuracy SD (%) | Cross-Entropy SD | Pairwise Disagree (%) | Pairwise Corr. | Ensemble Δ (%) |
|---|---|---|---|---|---|
| **CIFAR-10: ResNet-6** | | | | | |
| Parameter Initialization | 0.50 ± 0.04 | 0.0123 ± 0.0011 | 20.0 | 0.925 | 2.17 |
| All nondeterminism sources | 0.43 ± 0.03 | 0.0111 ± 0.0007 | 20.1 | 0.924 | 2.17 |
| Random Bit Change | 0.41 ± 0.02 | 0.0100 ± 0.0006 | 19.8 | 0.925 | 2.12 |
| Snapshot - All sources | 0.45 ± 0.03 | 0.0104 ± 0.0007 | 14.0 | 0.963 | 0.99 |
| **CIFAR-10: ResNet-10** | | | | | |
| Parameter Initialization | 0.23 ± 0.01 | 0.0064 ± 0.0004 | 13.7 | 0.912 | 2.13 |
| All nondeterminism sources | 0.23 ± 0.01 | 0.0068 ± 0.0005 | 13.6 | 0.911 | 2.13 |
| Random Bit Change | 0.25 ± 0.02 | 0.0068 ± 0.0005 | 13.5 | 0.913 | 2.08 |
| Snapshot - All sources | 0.22 ± 0.01 | 0.0047 ± 0.0003 | 8.8 | 0.962 | 0.96 |
| **CIFAR-10: ResNet-18** | | | | | |
| Parameter Initialization | 0.15 ± 0.02 | 0.0078 ± 0.0012 | 4.7 | 0.805 | 0.67 |
| All nondeterminism sources | 0.17 ± 0.02 | 0.0078 ± 0.0006 | 4.8 | 0.809 | 0.76 |
| Random Bit Change | 0.13 ± 0.01 | 0.0067 ± 0.0006 | 4.7 | 0.830 | 0.73 |
| Snapshot - All sources | 0.14 ± 0.01 | 0.0040 ± 0.0003 | 2.9 | 0.884 | 0.31 |
| **CIFAR-10: ShuffleNetv2-50%** | | | | | |
| Parameter Initialization | 0.22 ± 0.01 | 0.0119 ± 0.0008 | 8.4 | 0.696 | 1.38 |
| All nondeterminism sources | 0.22 ± 0.02 | 0.0131 ± 0.0008 | 8.4 | 0.692 | 1.40 |
| Random Bit Change | 0.21 ± 0.01 | 0.0115 ± 0.0007 | 8.3 | 0.695 | 1.36 |
| Snapshot - All sources | 0.18 ± 0.01 | 0.0067 ± 0.0005 | 5.0 | 0.930 | 0.52 |
| **CIFAR-10: VGG-11** | | | | | |
| Parameter Initialization | 0.20 ± 0.01 | 0.0068 ± 0.0005 | 6.6 | 0.807 | 0.91 |
| All nondeterminism sources | 0.18 ± 0.01 | 0.0069 ± 0.0005 | 6.6 | 0.806 | 0.94 |
| Random Bit Change | 0.16 ± 0.01 | 0.0060 ± 0.0004 | 6.5 | 0.811 | 0.89 |
| Snapshot - All sources | 0.13 ± 0.01 | 0.0041 ± 0.0003 | 4.1 | 0.914 | 0.39 |
| **MNIST** | | | | | |
| Parameter Initialization | 0.047 ± 0.0036 | 0.0025 ± 0.0002 | 0.54 | 0.941 | 0.064 |
| All nondeterminism sources | 0.046 ± 0.0032 | 0.0023 ± 0.0001 | 0.56 | 0.939 | 0.068 |
| Random Bit Change | 0.035 ± 0.0026 | 0.0011 ± 0.0001 | 0.30 | 0.989 | 0.011 |
| Snapshot - All sources | 0.050 ± 0.0031 | 0.0019 ± 0.0001 | 0.55 | 0.943 | 0.064 |

Table 6: Generalization experiments of nondeterminism and instability with other architectures on CIFAR-10 and additional experiments on a high-accuracy dataset (MNIST). Each row is computed from the statistics of 100 trained models (*i.e.* 2,400 models total for this table).

accuracy and cross-entropy comparable to an ensemble of 2, with other metrics comparable to those of a vanilla ensemble with even more members. By reducing the variability in model performance without increasing training time, this identifies accelerated model ensembling as a promising approach for further reducing the trained model variability and mitigating the effects of instability.

## 6 GENERALIZATION EXPERIMENTS

In this section we detail additional experiments showing our nondeterminism and instability claims generalize to other settings. For computational reasons (each result requires training the same model end-to-end $R = 100$ times), we restrict our attention to modestly-sized datasets and models.

On CIFAR-10, in addition to the ResNet-14 experimented with in Sec. 3 and Sec. 4 and the linear and single hidden layer models from Sec. 4.2, we experiment with 6-, 10-, and 18-layer ResNet variants, VGG-11 (Simonyan & Zisserman, 2014), and a 50%-capacity ShuffleNetv2 (Ma et al., 2018), allowing further exploration of the effect of model size and architecture on nondeterminism and instability. As shown in Table 6, the observations around instability and its relationship to nondeterminism generally hold for these architectures, with a close correspondence between the magnitude of effects for a random bit change and each of the four metrics considered.

We have also validated the benefits of accelerated ensembling via 5-checkpoint snapshot ensembles with all nondeterminism sources enabled for each of the models, where we find large and consistent reductions in run-to-run variability for pairwise disagreement, correlation, and changes from ensembling. For variability in performance (Accuracy and Cross-Entropy SD), we found increasing benefits as model size increased, where only for the smallest ResNet-6 was there no benefit.

We also perform experiments on MNIST (LeCun et al., 1998), allowing us to test whether our observations hold for tasks with very high accuracy — $99.14\%$ for our baseline model. As before, we find similar effects of nondeterminism for parameter initialization and all nondeterminism sources, including a comparable effect (albeit smaller) from a single random bit change, highlighting that the instability of training extends even to datasets where the goal is simpler and model performance is higher. Of note, though, is the relative smaller effect of a single bit change on pairwise metrics of diversity, further suggesting that the magnitude of instability might be at least partially related to the interplay of model architecture, capacity, and degree of overfitting. Snapshot ensembles, however, did not result in improvements on MNIST. While this is not particularly important in practice (since models on MNIST are fast to train), it is an interesting result, which we hypothesize is also related to the degree of overfitting (similar to ResNet-6 on CIFAR-10).

## 7 CONCLUSION

In this work, we have shown two surprising facts: First, though conventional wisdom is that run-to-run variability in model performance is primarily determined by random parameter initialization, many sources of nondeterminism actually result in similar levels of variability. Second, a key driver of this phenomenon is the instability of model optimization, in which changes on the order of $10^{-10}$ in a single weight at initialization can have as much effect as reinitializing all weights to completely random values. Model instability itself is a property of the depth of a model, with any model with more than single layer suffering from it when optimized with SGD. Beyond identifying and explaining this property of neural network optimization, we have also identified a promising direction for reducing the variability in model performance and representation without incurring the cost of training multiple copies of a model: accelerated model ensembling.

We hope that our work sheds light on a complex phenomenon that affects all deep learning researchers. Ultimately, we also hope that, aided by this understanding, the field is able to improve the stability of model training and accelerate progress in deep learning for all.

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

| Setting | Accuracy SD (%) | Cross-Entropy SD | Pairwise Disagree (%) | Pairwise Corr. | Ensemble Δ (%) |
|---|---|---|---|---|---|
| All Sources/.40/50 (*N=20*) | $0.31 \pm 0.04$ | $0.0074 \pm 0.0007$ | 11.4 | 0.922 | 1.54 |
| All Sources/.40/100 (*N=20*) | $0.26 \pm 0.04$ | $0.0072 \pm 0.0014$ | 10.8 | 0.909 | 1.71 |
| All Sources/.40/250 (*N=20*) | $0.19 \pm 0.03$ | $0.0060 \pm 0.0009$ | 10.7 | 0.889 | 1.78 |
| All Sources/.40/500 (*N=100*) | $0.26 \pm 0.02$ | $0.0075 \pm 0.0005$ | 10.7 | 0.871 | 1.82 |
| All Sources/.40/2000 (*N=20*) | $0.24 \pm 0.02$ | $0.0100 \pm 0.0014$ | 11.2 | 0.828 | 2.08 |
| Shuffle/.40/500 (*N=100*) | $0.25 \pm 0.02$ | $0.0087 \pm 0.0005$ | 10.6 | 0.871 | 1.81 |
| Shuffle/.20/500 (*N=100*) | $0.23 \pm 0.02$ | $0.0076 \pm 0.0005$ | 11.0 | 0.858 | 1.95 |
| Shuffle/.20/1000 (*N=100*) | $0.21 \pm 0.02$ | $0.0092 \pm 0.0005$ | 11.1 | 0.837 | 2.02 |
| Shuffle/.10/500 (*N=100*) | $0.20 \pm 0.01$ | $0.0080 \pm 0.0006$ | 11.6 | 0.845 | 2.08 |
| Shuffle/.10/2000 (*N=100*) | $0.24 \pm 0.02$ | $0.0106 \pm 0.0007$ | 11.6 | 0.801 | 2.19 |
| Param. Init/.40/500 (*N=100*) | $0.22 \pm 0.02$ | $0.0080 \pm 0.0005$ | 10.7 | 0.872 | 1.82 |
| Param. Init/.20/500 (*N=100*) | $0.23 \pm 0.02$ | $0.0089 \pm 0.0005$ | 11.0 | 0.859 | 1.97 |
| Param. Init/.20/1000 (*N=100*) | $0.25 \pm 0.02$ | $0.0102 \pm 0.0007$ | 11.1 | 0.836 | 2.06 |
| Param. Init/.10/500 (*N=100*) | $0.26 \pm 0.02$ | $0.0087 \pm 0.0006$ | 11.7 | 0.844 | 2.13 |
| Param. Init/.10/2000 (*N=100*) | $0.22 \pm 0.01$ | $0.0099 \pm 0.0008$ | 11.6 | 0.800 | 2.18 |
| All Sources/.40/200 (*N=100*) | $0.23 \pm 0.02$ | $0.0080 \pm 0.0005$ | 10.6 | 0.895 | 1.75 |
| Random Bit/.40/200 (*N=100*) | $0.21 \pm 0.01$ | $0.0071 \pm 0.0004$ | 10.3 | 0.897 | 1.70 |
| Snapshot/All Sources/.40/200 (*N=100*) | $0.21 \pm 0.01$ | $0.0046 \pm 0.0003$ | 6.6 | 0.963 | 0.68 |

Table 7: Experiments varying the learning rate and number of epochs for ResNet-14 on CIFAR-10. In each row, the experimental setting is abbreviated by [sources of nondeterminism]/[maximum learning rate]/[number of epochs] *N*=[number of models trained], with the exception of the last row, which is a Snapshot ensemble but otherwise follows the same format.

## A   APPROACHES THAT DON'T REDUCE INSTABILITY

In the process of finding an approach that reduces run-to-run variability of models (Sec. 5), we experimented with many approaches which all failed to make a dent in improving variability and stability. For the benefit of the field, here we provide our experiences with these approaches which did not succeed in improving stability, despite the intuitive arguments for why they might help.

LEARNING RATE AND DURATION OF TRAINING.    Noticing that the effects of nondeterminism seemed to accumulate during the course of training (Fig. 2), it seemed reasonable that varying the learning rate or duration of training might have an effect. However, varying the duration of training from anywhere between 50 and 2,000 epochs on CIFAR-10 all produced models with a similar variance in performance as the results in the rest of this work (which used 500 epochs), even though the absolute performance differed by up to $\sim 2\%$.

We show these results in Table 7. In general, increasing the number of epochs or changing the learning rate did not change the variability in performance (Accuracy SD; Cross-Entropy SD) much, with only a very slight increase in variability as the number of epochs grew to extremely large values (*i.e.* 2,000 epochs). There were slightly larger changes in pairwise representation-based metrics, where training longer again increased run-to-run variability. However, none of these attempts actually reduced variability; they only served to potentially make it larger.

As part of these experiments, we also verified the instability behavior with only 200 epochs of training and the effectiveness of accelerated ensembling techniques ("Snapshot") with this reduced training time.

USING A DIFFERENT OPTIMIZER.    Since instability and nondeterminism are both a property of optimization, it is conceivable that use of a different optimizer might be able to lessen the degree of instability in model training. We experimented SGD with various values of momentum, ranging from 0 for pure SGD to 0.999 for a momentum-driven optimizer, but none succeeded in reduce instability. In addition, we experimented with the Adam (Kingma & Ba, 2014), as a representative of the class of adaptive learning rate algorithms, but this, too, had no effect on stability.

AGGRESSIVE STOCHASTIC WEIGHT AVERAGING.    Inspired by the success found by Madhyastha & Jain (2019), we tried using Aggressive Stochastic Weight Averaging (ASWA), a variant of SWA (Izmailov et al., 2018). However, we could not get the model to converge to a reasonable degree of

| Nondeterminism Source | Accuracy SD (%) | Cross-Entropy SD | Pairwise Disagree (%) | Pairwise Corr. | Ensemble $\Delta$ (%) |
|---|---|---|---|---|---|
| **CIFAR-10: Linear model** | | | | | |
| Parameter Initialization | $0.03 \pm 2e\text{-}3$ | $0.0002 \pm 1e\text{-}5$ | 0.5 | 0.997 | -4e-3 |
| All nondeterminism sources | $0.10 \pm 0.01$ | $0.0007 \pm 4e\text{-}5$ | 5.5 | 0.996 | 0.06 |
| Random bit change | $0.00 \pm 0.00$ | $2e\text{-}8 \pm 1e\text{-}9$ | 0.0 | 1.000 | 0.00 |
| **CIFAR-10: One hidden layer (fully-connected)** | | | | | |
| Parameter Initialization | $0.31 \pm 0.02$ | $0.0054 \pm 0.0004$ | 24.2 | 0.941 | 1.38 |
| All nondeterminism sources | $0.30 \pm 0.02$ | $0.0057 \pm 0.0005$ | 24.9 | 0.937 | 1.49 |
| Random bit change | $0.28 \pm 0.02$ | $0.0055 \pm 0.0005$ | 23.4 | 0.945 | 1.32 |
| **CIFAR-10: One hidden layer (convolutional)** | | | | | |
| Parameter Initialization | $0.26 \pm 0.02$ | $0.0043 \pm 0.0003$ | 12.5 | 0.974 | 0.64 |
| All nondeterminism sources | $0.22 \pm 0.01$ | $0.0044 \pm 0.0003$ | 12.7 | 0.973 | 0.68 |
| Random bit change | $0.14 \pm 0.01$ | $0.0024 \pm 0.0002$ | 6.4 | 0.993 | 0.18 |

Table 8: Linear and 2-layer experiments on CIFAR-10

performance with the original formulation due to update sizes that decreased too rapidly, and though we were able to modify it to converge successfully, the output variance remained as high as the other models.

GRADIENT CLIPPING.    With the intuition that instability might be caused by spurious gradients of large magnitude, we experimented with clipping the norm of gradients (using `pytorch`'s implementation of `torch.nn.utils.clip_grad.clip_grad_norm_`. Like other approaches, though, this had no effect on model variability.

WEIGHT AUGMENTATION.    A very experimental approach, to reduce instability we experimented with taking an averaged gradient around the current set of parameters at each step, approximated by sampling a random weight offset before doing a forward or backward pass through the model. Intuitively, this would encourage optimization to not be too sensitive to the current value of weights; however, in practice this simply didn't affect the variance or stability of the model.

## B    ADDITIONAL LINEAR AND 2-LAYER RESULTS

In Table 8 we include results for linear networks and 2-layer networks (1 hidden layer) on CIFAR-10, omitted from the main text due to space constraints. Of note in these results is the relative smaller effect of a single bit change for a 2-layer network where the hidden layer is convolutional — still much larger than for the linear model, but significantly smaller than for any other non-linear model. This suggests a similar effect to what was previously observed on MNIST (Table 6) in that degree of instability might be related to the interplay of model, dataset, and the degree of overfitting.

## C    IMPACT OF RANDOM BIT CHANGE OVER TIME

In Fig. 3 we plot the effect of a random bit change for a linear and single on CIFAR-10, illustrating the effect of instability as described in Sec. 4. In the first few epochs of training, we observe that the standard deviation and range of cross-entropy for the model with one hidden layer quickly grows, only eventually decreasing much later in training as the model's parameters converges toward their final values. On the other hand, for linear models, the standard deviation consistently remains 5 or more orders of magnitude lower throughout training.

## D    SUBTLETIES IN EVALUATION

While we have done our best to present our experimental protocol as plainly as possible, one subtlety arises when interpreting results. First, recall that in our example from Sec. 3.1, testing the effects

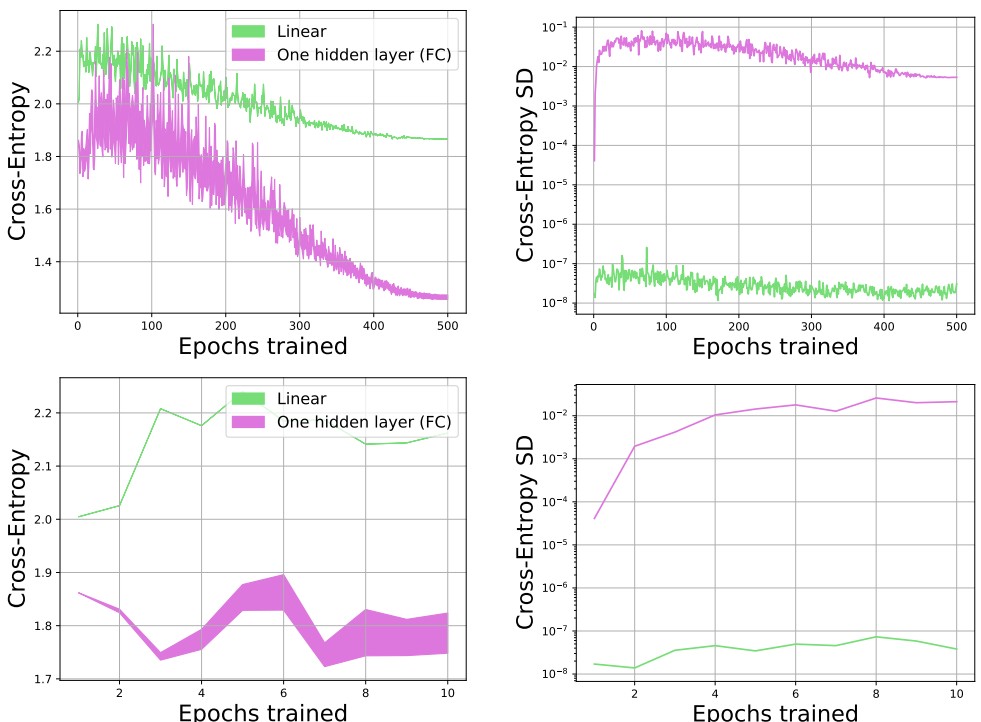

Figure 3: The impact of a random bit change during initialization for linear models vs 2-layer models with a single fully-connected hidden layer, where row 1 considers the full 500 epochs of training, and row 2 zooms in on the first 10 epochs. The left column of each row gives the range of cross-entropy values for 100 models in the middle 95th percentile of cross-entropy, plotted at each epoch. The right column of each row presents the standard deviation of these models.

of random initialization corresponded to training models for $(S_1, S_2, S_3) \in \{(i, 1, 1)\}_{i=1}^{R}$, where $S_1$ is the seed for random initialization, $S_2$ is the seed for training data shuffling, and $S_3$ is set to 1 to indicate the deterministic mode for cuDNN. The subtlety arises in that the resulting distribution of $(S_1, S_2, S_3) \in \{(i, 1, 1)\}_{i=1}^{R}$ is not necessarily the same as the distribution where $S_2$ is set to a different arbitrary constant value, *e.g.* $S_2 = 2$. Due to this, there may be minor discrepancies when comparing the diversity in performance between two different sources of nondeterminism (albeit nothing likely to change general conclusions). Combined with the natural sampling variability implicit in only training a finite number of models, this can lead to paradoxical results such as the standard deviation for a particular metric being slightly higher for a random bit change as compared to an entirely different random parameter initialization. While we have separately validated that the general conclusions of our results hold when varying a few of these constant factors (*i.e.* running experiments where $S_2$ is set to 2 and 3, in this example), it is difficult to resolve the discrepancy entirely without models according to the full cross-product of random seeds, which is prohibitive due to the exponential number of such models.

