# OpenReview forum: "On Nondeterminism and Instability in Neural Network Optimization"
_ICLR.cc/2021/Conference — Reject_

### Official Review · AnonReviewer1 · 2020-10-14
**An interesting experimental analysis about neural network instability, but that which is not completely  convincing**

**Rating:** 5
**Confidence:** 3

**Review:**

Summary of the paper:
The authors analyze the effect of sources of uncertainty on neural network performance. In particular, they consider the effect of parameter initialization, data shuffling, data augmentation, regularization, and the choice of deep learning libraries on network performance and show that all of these aspects have similar effects. Furthermore, the authors claim that these sources of uncertainty are all dependent on network weights, with even small changes in network weights drastically affecting the network’s performance.  The authors use statistical measures such correlation between model predictions, change in performance with and without ensemble game models, and a state of the art method to characterize the functional behavior. Results are reported for image classification and language modeling .

Positives:
1. The paper offers some interesting revelations such as : all sources of uncertainty have similar effects, which is surprising as the authors note, and hence a valuable insight.
2. The problem is well motivated, and the presentation is mostly clear.
3. Experiments have been conducted on diverse domains (image and language) to demonstrate the effectiveness of the proposed method.
Concerns:
1. Technical sophistication:
-As I understand, the goal is to be able to quantify the effect of various sources of non determinism on performance. Fundamentally, this seems like a causal attribution problem. While correlation based metrics can offer insight, it is not sufficient enough to establish causal claims.
-Moreover, the different sources of non determinism may be influencing each other. The authors in one of their protocols study the effect of each of these in a rather independent fashion, which makes it hard to estimate the influence of one source on another if any. It is therefore necessary to analyze all possible combinations of source variations for the conclusions made to hold true.
-Also, if one thinks of the problem as that of causal inference and imagines a DAG whose nodes are various sources of non determinism, then the sources which need to be controlled for will be provided by adjustment formulas. This is more concrete than just controlling for few sources as the authors propose because it is not guaranteed to remove all spurious correlations.
-As one of the ways to address the problem, the authors suggest leveraging snapshot ensembles. It is not clear how the non determinism that can arise in this model itself (e.g. choice of samples in the ensembles) does not affect the performance.

2. Novelty
- One of the methods in the protocols is a state of the art method for functional analysis of neural networks, and the other two are are common measures. So, the contribution from a protocol perspective is not significantly novel.

 Minor Comment:
It would help to define non determinism formally early on in the paper. While the paper provides sufficient motivation and later on describes the various sources, it still helps to define the term in one sentence or two.

Question to authors:
- Has the ordering of changing sources sequentially and simultaneously been analyzed?
- How are the sources of non determinism in the snapper ensembles overcome? Or are there no such sources?

Overall comments:
The paper offers some interesting insights via experiments. Some aspects about the protocol metrics are intuitive as the authors explain, despite this, a theoretical analysis to back the experimental findings would have made the paper stronger. This is because it is hard to be convinced that the process of attribution can be established entirely based on statistical measures. The relationships between various sources and the graphs describing their dependencies have to be analyzed in determining the sources that need to be adjusted for in determining causal effects.

---

> ### Author Response · Authors · 2020-11-20
> **Thank you for your comments**
>
> Thank you for your comments and drawing an interesting connection to causal attribution.
>
> Causal attribution: Although studying the problem from a causal attribution framework is interesting and would strengthen this or future research, examining all possible combinations of sources of nondeterminism simply isn't tractable due to the exponential blowup in the number of such combinations (hopefully we are interpreting your suggestion correctly to examine the full cross-product of seeds for each nondeterminism source, which corresponds to ~2 million independent model trainings for the seeds in this work). We did not mean for our conclusions to be interpreted in the sense of causal inference and will clarify and soften some of the language in the revision to make it less ambiguous.
>
> Snapshot determinism: The choice of snapshot checkpoints is deterministic -- for example, when using a total of 500 epochs of training and 5 members in the snapshot ensemble, the ensemble is comprised of the models after training for 100, 200, 300, 400, and 500 epochs. We will clarify this (and other motivation for snapshot ensembles) in the revision.
>
> Novelty of variability measures: Since we are the first to study the problem of nondeterminism on model optimization in this level of detail, we aimed to have very intuitive, easy-to-understand measures, only using more complicated metrics (e.g. CKA) when they made sense. Our novelty lies in being the first to study the problem and reveal insights on nondeterminism and instablity rather than having a particularly complex approach.
>
> Defining non-determinism: Thank you for the suggestion! We will incorporate this in our revision.
>
> Changing nondeterminism sequentially and simultaneously: We vary a subset of sources simultaneously (e.g. "Data Shuffling + cuDNN", "Data Shuffling + Aug. + cuDNN", and "All nondeterminism sources" in Table 1), and vary sources sequentially (changing seeds across time during training) in Sec. 3.4. However, we are not sure if we are interpreting the reviewer correctly, and any clarification would be much appreciated so that we can improve our work.

---

### Official Review · AnonReviewer4 · 2020-10-26
**The empirical evidences are interesting, paper can be strengthened by more discussion on prior work and theoretical insights.**

**Rating:** 6
**Confidence:** 4

**Review:**

The paper investigates the effect of nondeterminism and stability in Neural Networks (NNs) for supervised learning tasks in a systematic manner. The paper is very well-written. All the steps towards the claims of the paper are clearly stated. The empirical analysis is systematic and the two main results are thought provoking and interesting: 1) Different sources of nondeterminism (such as random initialization, data augmentation, data shuffling, etc.) causes similar levels of variability (based on standard deviation and correlation metrics), and 2) Changes in the optimization even in the order of 10^-10 in a single weight can have same variability level as changing the random seed entirely. The paper also validates that a prior work called Snapshot Ensembles (to some degree) resolves the instability problem in NNs.

My only concern is on the significance of the method given the prior work on Snapshot Ensembles. Although the work is independently interesting and opens up new questions for the field it also seems to be a great motivational section to develop Snapshot Ensembles which has already been published. Are there any modifications in the SnapShot Ensembles that would result in better stability results? I think the paper would be strengthened by a discussion on how the prior work designed for a different objective is working to reduce stability and how the stability issues has not been acknowledged in the Snapshot Ensembles paper.

Some questions: given such robustness in the level of variability across different sources of nondeterminism, is it possible to predict the level of variability from the data, architecture, etc.? Is there anything more fundamental about the particular value of variability that all the different sources of nondeterminism concentrate around? Can this be formally characterized?

** after rebuttal: thanks for the efforts in updating the paper. I will stick with my score.

---

> ### Author Response · Authors · 2020-11-20
> **Thank you for your comments!**
>
> Thank you for your comments and appreciation of our analyses. We are glad that you share our enthusiasm for our main results!
>
> Developing accelerated ensembling approaches further: We couldn't agree with you more that this is worth looking into, as it is (so far) the only promising approach for reducing variability, and this is definitely in scope for our next future work. The majority of our paper is on analyzing the effects of nondeterminism and establishing the role of instability in its behavior. The experiments with snapshot ensembles are meant to show that, despite instability, and despite many intuitive approaches not reducing the variance of performance (Appendix A), progress is still possible, and that accelerating ensembling methods are a promising direction for further research. We will add more motivation for why snapshot ensembles help reduce model variability in our next revision.
>
> Predicting variability from data, architecture, etc.: This is also an interesting avenue for future work. While it's clear that some amount of variability can be predicted (e.g. that linear models will always have very low variability), it's unclear what general patterns will emerge, similar to how it may be difficult to predict model performance itself (i.e. not the variability) from these factors. There is also an interesting connection here to work in neural architecture search, which typically looks to maximize the first moment of model performance, while this would consider the second moment.

---

### Official Review · AnonReviewer2 · 2020-10-28
**Extremely interesting, experiments and analysis on optimization instability can be improved**

**Rating:** 6
**Confidence:** 4

**Review:**

### Summary:

This paper sheds light on the impact of nondeterminism to the run-to-run variability of neural network performance---a situation many people using neural networks have experienced. The authors establish an experimental strategy to analyze the different sources of nondeterminism. Some sources of nondeterminism are parameter initialization, data shuffling, data augmentation, regularization and cuDNN.

The authors make the surprising discovery that each source of nondeterminism results in an equal amount of variability and model diversity. By modifying weights by a single bit, they experimentally demonstrate that an inherent instability in the neural network optimization procedure is the main reason. They show methods such as snapshot ensembling can reduce the observed variability.


########################################

### Strong points:

The discovery that each nondeterminism source has a similar effect is novel and is not in the literature as far as I know. Such results are intriguing, unexpected and useful.

The experimental methodology used is well-explained and fair.

Linking all the nondeterminism to a change of one bit in model weights is interesting and successfully highlights the instability and sensitivity of neural network optimization.


########################################

### Weak points:

Please can the authors clarify the takeaways of section 4.2? At the moment the novelty or surprise is not entirely clear. If the single linear layer problem is convex, it is expected that a single bit initialization perturbation still leads to the global minimum. Due to nonconvexity, the one hidden layer networks can have different minima and so there is more variability. But how does the extent of the variability change with depth?

Please can the authors elaborate on the connection between the ensembling solution and how it prevents optimization instability which was identified as a root cause? Ensembling methods in general will help variability but there is no change in the optimization and training process and ensembling gains are due to other reasons.

While it is understandable that there is significant compute time required to do multiple runs, the networks used are much smaller than current state-of-the-art networks. For example, a WideResNet-28x10 can give approximately 95% test accuracy for CIFAR-10 and has a much larger capacity than the ResNet-14. The paper would be strengthened by having experiments on popular benchmark state-of-the-art networks (maybe fewer runs and not as many nondeterminism sources as in Table 1). This is especially relevant due to the huge gulf in the number of trainable parameters in state-of-the-art networks which means potentially very different functions can be learned run-to-run---do the trends in Table 6 carry through for big networks?

Why were 500 epochs used for the CIFAR experiments? 500 epochs are plenty for training the ResNets for CIFAR-10 classification and is much larger than what is common in the literature. Furthermore, with five snapshots, this is like having five models trained for 100 epochs each which on its own is enough (or almost enough) for a single model. Do the authors have a more realistic set of experiments (such as a total of 200 epochs and four to five snapshots)?


########################################

### Recommendation:

Overall I lean towards rejection. I think the results (such as Table 1) are extremely interesting and should be known in the community, however, the current analysis of the optimization instability should be developed. Furthermore, the experiments could be improved as described elsewhere in this review.


########################################

### Additional questions and clarifications that will help assessment:

Please address and clarify my above questions and queries.

In the experiment of Figure 2, is the cosine learning rate decay used from epoch 0? If so at later epochs when the nondeterminism is activated, the learning rate is lower and the model cannot explore and change as much and so reduction in variability may in part be due to the learning rate.

Section 6 mentions that there are 18 layer ResNet experiments. Please can the authors include these?


########################################

### Additional feedback that does not necessarily impact recommendation:

Typo:

\- Page 7, first paragraph ‘in this was with only’

\- Appendix B says that Table 7 is for MNIST, but the table says CIFAR


######### After author response #########

I thank the authors for their responses and updates to the manuscript. While I still feel that the connections between optimization instability and the observed phenomena could be developed further, the updates strengthen the paper and the experimental results are extremely interesting. I have increased my score for these reasons.

---

> ### Author Response · Authors · 2020-11-20
> **Thank you for the review! Response to comments + additional experiments**
>
> We appreciate your thorough reading and helpful comments of our work. Here are the answers to your questions:
>
> Sec 4.2: What we found is stronger than linear models converging to the same optimum -- rather, the entire optimization trajectory itself remains nearly unperturbed. We will add a figure to the Appendix to further demonstrate this in our revision within the next week. In terms of how variability changes with depth, this can be determined by comparing our results for a subset of the architectures considered in our work (linear, two-layer, ResNet-6, ResNet-10, and ResNet-14, with an additional ResNet-18 to be added).
>
> Ensembling and optimization instability: As you correctly point out, ensembling by itself doesn't mitigate instability in optimization, but ensembled models themselves have reduced variance or diversity since they are composed of multiple submodels, averaging the predictions of each of its members. This averaging is what drives the variance reduction. Thank you for pointing out this opportunity to add clarity in our writing, which we will add in our revision.
>
> Larger networks: We will add a wide ResNet-18 model to the revision, which has an accuracy of ~94.9% on CIFAR-10 and shows a similar trend in results.
>
> Number of epochs: This is a very good question! We originally chose 500 epochs for all of our experiments out of caution, making sure that models had enough time to converge (though as noted in the appendix, changing the training duration did not have much effect on model variability). Nonetheless, we do see your point that this could give an unfair advantage to the snapshot approach in reducing variability. Thus, we will add new experiments with 200-epoch training to our revision, which we have found show the same trend as the others despite the reduced training time.
>
> ResNet-18: Yes, we will add these experiments to our revision.
>
> Learning rate: Good question -- yes, a cosine learning rate decay was used starting from the beginning (after three epochs of learning rate warmup). We have since performed a handful of experiments varying the learning rate (and optionally varying the number of training epochs to compensate), which we will add to the Appendix in the revision. In short, the maximum learning rate does not appear to have a significant effect on model variability, leading us to believe that it is not the primary driver of this curious effect.
>
> Typos: Thank you for pointing these out! We will fix these in our revision.

---

### Official Review · AnonReviewer3 · 2020-10-29

**Rating:** 5
**Confidence:** 3

**Review:**

This paper discusses nondeterminism and instability in neural network optimization.  The authors establish an experimental protocol to understand the effect of optimization nondeterminism on model diversity, and study the independent effect of different sources of nondeterminism.

Pros:
1. Show that different sources of nondeterminism give similar level of variability effect on the final accuracy and loss.
2. The source of the above phenomenon comes from model optimization instability.
3. Provide one possible direction for reducing the variability.

Cons:
1. The accelerated model ensembling is from another work and using ensemble to reduce variability is quite intuitive.
2. More experiments need to be done on larger datasets to further demonstrate the findings of this work.
3. More novelty is needed for this work to be published in ICLR.

---

> ### Author Response · Authors · 2020-11-20
> **Thank you for your comments!**
>
> Thank you for reading and providing comments on our work.
>
> Scale/Novelty: With all due respect, we think that the novelty of the paper lies in the impact of researching this problem -- the problem we study affects nearly all of deep learning, and our work is the first to study it in depth. Doing so necessarily requires training hundreds of models for a single experiment (e.g. 700 trained models for Table 1 alone). Repeating this on a larger dataset (e.g. ImageNet), even in the optimistic scenario of model training taking only 24 hours on a single GPU to converge, would take roughly 2 years of GPU time. We agree that such research would be beneficial to the community in making our findings even more robust. However, doing so is simply only possible for very few research labs with very large computational resources. By choosing to study the problem with more tractable models and datasets, we have been able to conduct detailed analyses on nondeterminism during training and discover two highly non-intuitive facts: 1) that each source of nondeterminism results in similar levels of final model variability and diversity, and 2) that instability in optimization offers an explanation for this.
>
> Ensembling intuition: While we thank you for your comment, we have a different opinion. Though some may find it intuitive that ensembling can reduce model variability, it is typically not the first choice one would make (i.e. training multiple model replicas and taking the average). It is also unclear that accelerated model ensembling strategies would necessarily maintain the advantage of ensembling in reducing variance. In fact, many approaches exist with at least some intuition backing them up, yet fail to work (Appendix A).

---

### Decision · Program_Chairs · 2021-01-07
**Final Decision**

**Decision:**

Reject

**Comment:**

This paper investigates the topic of nondeterminism and instability in neural network optimization. The reviewers found the results on different sources of nondeterminism particularly interesting and relevant. The experiments are carried on both language and also vision, which strengthens the findings. Concerns were raised about the use of smaller non-standard models, which were somewhat mitigated by the addition of Resnet-18 experiments on CIFAR. The reviewers also noted that the measures used in the experimental protocol were already present in the literature, and that the proposed mitigation strategy is from another work. Furthermore, R2 also found that the optimization instability section should be more developed. The paper should be resubmitted with an improved discussion of related works and more developed section on instability as suggested by the reviewers.